# Unveiling the Nutritional Quality of Terrestrial Animal Source Foods by Species and Characteristics of Livestock Systems

**DOI:** 10.3390/nu16193346

**Published:** 2024-10-02

**Authors:** Ana María Rueda García, Patrizia Fracassi, Beate D. Scherf, Manon Hamon, Lora Iannotti

**Affiliations:** 1Food and Agriculture Organization of the United Nations, 00153 Rome, Italy; ana.ruedagarcia@fao.org (A.M.R.G.); patrizia.fracassi@fao.org (P.F.); beate.scherf@fao.org (B.D.S.); manon.hamon@fao.org (M.H.); 2E3 Nutrition Lab, Washington University in St. Louis, St. Louis, MO 63130, USA

**Keywords:** animal source foods, terrestrial animal source food, TASFs, nutrient, macronutrient, micronutrient, food composition, food quality, livestock, livestock systems

## Abstract

**Background.** It is well-established that a range of macronutrients, micronutrients and bioactive compounds found in animal-source foods play unique and important roles in human health as part of a healthy diet. **Methods.** This narrative review focuses on terrestrial animal source foods (TASFs). It particularly analyzes five groups: poultry eggs, milk, unprocessed meat, foods from hunting and wildlife farming, and insects. The objectives were as follows: (1) examine the nutrient composition of TASFs within and across livestock species, drawing on the country and regional food composition databases; (2) analyze the influence of intrinsic animal characteristics and production practices on TASF nutritional quality. **Results.** TASFs are rich in high-quality proteins and fats, as well as micronutrients such as vitamin B12, iron or zinc. This study found differences in the nutritional quality of TASFs by livestock species and animal products, as well as by characteristics of livestock production systems. Our findings suggest that there may be public health opportunities by diversifying TASF consumption across species and improving certain aspects of the production systems to provide products that are both more sustainable and of higher quality. **Conclusions.** Future research should adopt a more holistic approach to examining the food matrix and the dietary patterns that influence TASF digestibility. It is necessary to include meat from hunting and wildlife farming and insects in global food composition databases, as limited literature was found. In addition, scarce research focuses on low- and middle-income countries, highlighting the need for further exploration of TASF food composition analysis and how intrinsic animal characteristics and livestock production system characteristics impact their nutritional value.

## 1. Introduction

Animal-source foods, including eggs, fish, milk and dairy products, and meat, can play a significant role in nutrition and human health as part of a healthy diet [1,2,3,4]. In particular, this is especially important during key life stages, including pregnancy and lactation, childhood, adolescence and older age [1,2,5,6,7,8]. This narrative review focuses on terrestrial animal source foods (TASFs). The term TASFs includes poultry eggs, milk and dairy products, meat and meat products, foods from hunting and wildlife farming, and insects, which are a heterogeneous group in terms of nutrient content [2]. The review particularly focuses on poultry eggs, milk, unprocessed meat, foods from hunting and wildlife farming, and insects.

There is consistent and well-established evidence on the importance of certain TASFs and their nutrients for human health [1,2,3,5,6,7,8]. However, the lack of nutrient characterization for some TASFs across species highlights a significant gap in the research. In addition, with a focus on TASFs, there is a need to synthesize food data composition from different databases to facilitate a comprehensive and global understanding of nutrient analysis. This lack of information emphasizes the urge for further studies that allow a more holistic picture of TASF composition within and across livestock species.

Intrinsic animal characteristics, husbandry practices and other specifics of livestock production systems influence TASF quality. These characteristics cover parameters such as breed, genetic traits, sex and age. Livestock husbandry practices, along with other elements of the production systems, include methods and measures applied by farmers and pastoralists. This encompasses a range of aspects, such as feeding, reproductive management, housing, and health and welfare care from birth to slaughter [2]. Understanding how nutrient variability by livestock characteristics and production systems influence TASF quality is crucial to developing evidence-based statements on nutrient content and their potential effects on human health. It will allow policy makers to establish more precise recommendations and consumers to make better-informed choices when buying TASFs.

The objectives of this narrative review were as follows: (1) examine the nutrient composition of TASF groups within and across livestock species drawing on the country and regional food composition databases; (2) analyze the influence of animal characteristics and livestock production systems characteristics on TASF nutritional quality.

## 2. Materials and Methods

The following review is based on a literature search using three databases: EBSCOHost, PubMed and ScienceDirect. It covers studies that examine the nutrients found in TASFs and the composition of TASFs by livestock species. The complete search strategy is available in Appendix A from the Appendix A. To compare the nutrient composition of TASFs across multiple species and regions, we built a unique TASF composition database. The following databases were included: ASEAN Food Composition Database (Indonesia, Malaysia, Philippines, Singapore, Thailand and Vietnam), Australian Food Composition Database (Australia and New Zealand), Tabla de Composición de Alimentos Colombianos (Colombia), Frida (Denmark), FoodData Central (United States of America) and FAO/INFOODS (Western Africa). Nutrient composition from livestock species not found in food databases was included in the literature [9,10].

We included nutrients with sufficient data available: energy, protein, fat, carbohydrate, vitamin A, riboflavin (vitamin B2), cobalamin (vitamin B12), calcium, iron and zinc. The average of these nutrients are shown by TASFs. For more information about the creation and standardization of the tables, see Appendix A in the Appendix A. The review includes insects despite their usual exclusion from databases, as there has been a growing body of literature on the topic. Furthermore, it incorporates foods from hunting and wildlife farming as a separate group due to their common characteristics. Aquatic foods, foods that comprise TASFs as ingredients and fortified TASFs were not included as they were beyond the scope of the review. To ensure consistency, products derived from eggs, meat, dairy and insects were excluded. To maintain a uniform variable dataset allowing comparisons, the study covered generally raw TASFs. This helps to minimize confounding factors that might arise from the varying processing methods.

## 3. Terrestrial Animal Source Foods Nutrient Composition

Below, we summarize the findings of our analyses on the nutrient composition of various TASF groups within and across livestock species. For each group, we use a similar structure covering first macronutrients followed by micronutrients.

### 3.1. Poultry Eggs

The most commonly consumed type of eggs in our food systems is chicken eggs. However, many societies also incorporate eggs from other poultry species into their diets (for example, duck, goose, quail and turkey) (Table 1). Poultry eggs present a variety of different macronutrients and micronutrients but have notably high content of high-quality proteins, essential fatty acids, Docosahexaenoic acid (DHA), choline, riboflavin (vitamin B2), vitamin B12 and selenium [11]. The size and color of eggs differ depending on the species. For example, goose or turkey eggs are 2–2.5 times larger than chicken eggs [12].

Eggs are composed, on average, of 11% of shell, 58% of albumin (or egg white) and 31% of yolk [13]. The existing research on the nutrient profile of eggs frequently distinguishes between the nutrient concentrations of eggs’ yolk and white. These components develop at different stages of egg production. The yolk, which is mainly nutritive, is generated from hepatic tissue, and the white, which primarily serves for defense and consists mainly of water and a major portion of essential amino acids, is secreted in the oviduct [14]. The proteins and lipids of the vitelline membrane in eggs separate the egg yolk from the white.

Eggs have a combination of all essential amino acids, which is regarded as a reference standard against which other proteins are compared [15]. The concentration of proteins is relatively consistent across poultry species, with quail eggs showing the highest average and turkey eggs the lowest in our analysis (Table 1). The proteins in eggs are recognized for their high digestibility. This digestibility is enhanced further via heating, which denatures certain structural proteins [16]. The primary proteins found in egg whites are ovalbumin and ovomucoid. These are proteinase inhibitors that remain stable under thermal heating. Egg white also contains avidin, cystatin, lysozyme, ovoflavin, ovoglobulin, ovotransferrin and peptides, which are antibacterial proteins and likely contribute to human nutrition [17]. Among the components of egg yolks, a combination of low- and high-density lipoproteins, livetins and phosvitins are noted. Phosvitins, in particular, have been shown to have an influence on iron absorption [18]. Carbohydrates in eggs are low.

Different poultry species present a wide variety of fatty acids in their egg composition. The concentration of lipids is substantially located in the egg yolk, with a much lower concentration in the egg white [16]. As for the egg yolk, the main fatty acids in egg white are oleic, palmitic, arachidonic, linoleic and stearic acids. However, lipids are one of the main components of egg yolk, constituting 30–36% of it [12]. On average, chicken eggs contain fewer lipids than the other poultry species analyzed (duck, goose, quail and turkey) (Table 1). One study revealed that eggs have a higher proportion of unsaturated fats relative to saturated fats in comparison with other TASFs, which offer health benefits [16]. Additionally, eggs have relatively high cholesterol concentrations in comparison to other foods, although there is some uncertainty in the evidence base, which correlates the intake levels with plasma cholesterol [19].

Eggs are a major dietary source of choline and contain both lipid-soluble and water-soluble forms of the nutrient. Choline can be synthesized by the liver in limited quantities; however, the primary source is obtained through dietary intake [20]. Eggs also contain vitamin B12 (eggs from different poultry species can supply over 50% of age- and sex-specific recommended nutrient intake in a single egg), as well as considerable concentrations of additional B vitamins. From the species analyzed, duck eggs have the highest content of vitamin A (Table 1). Egg white has a lower content of minerals than egg yolks, though this can vary among species. Turkey eggs generally have higher concentrations of important minerals, including calcium, iron and zinc, compared to eggs from other birds, followed by quail eggs (Table 1). However, the evidence related to the effect of dietary egg intake on human mineral nutrition remains equivocal [2].

### 3.2. Milk

Cattle is currently the main animal species associated with milk and dairy products. However, generally, milk and dairy products from numerous livestock species are consumed every day, as they are an important source of nutrition (Table 2). The composition of these products varies among livestock species (Table 2). Understanding the different nutrient profiles, such as the content of lactose or proteins, of milk and dairy products among species can also facilitate their consumption for individuals with intolerances or allergies. The major components in milk are protein, lipids, lactose and minerals [21]. Protein content per 100 g is found in high concentrations in the milk of reindeer, followed by mithan, sheep and alpaca (Table 2). According to one analysis, the concentrations of protein in plant-based alternatives (coconut, hemp, nuts, oats, rice and soybean) approximately contained an estimated 48% of that in cow milk [22].

Milk lipids mainly consist of triacylglycerols (98%) and minimal quantities of diacylglycerol (2%), cholesterol (<0.5%), phospholipids (1%) and free fatty acids (0.1%) [23]. Cow milk contains almost 70% saturated fatty acids and 30% unsaturated fatty acids. The presence of oleic acid, conjugated linoleic acid and omega-3 fatty acids in cow milk may contribute to beneficial human health outcomes [24]. Fat content per 100 g is found in high concentrations in the milk of reindeer, followed by mithan, buffalo and yak. In contrast, the milk of mare and donkey contains the lowest levels of protein and fat (Table 2). A review highlighted that, on average, milk from mare and donkey contains lower levels of saturated fatty acids and higher content of polyunsaturated fatty acids (PUFA) compared to milk from ruminants [21]. While ruminant milk has higher levels of monounsaturated fatty acids (MUFA), higher ratio of omega 6 to omega 3 and cholesterol in comparison to mare and donkey milk [21].

Concerning micronutrients, milk is particularly noted for its high mineral content and, in particular, its high bioavailability of calcium. The calcium is bound to casein in micellar form and to whey proteins and inorganic salts. Cow milk has the highest content of calcium per 100 g of milk (Table 2). In addition to magnesium, phosphorous and potassium, which are macrominerals concentrated in milk, it is also rich in the microminerals selenium and zinc. Iodine is present in cow milk [2]. The content of fat-soluble vitamins varies across milk and dairy products due to their fat concentration (whole, low-fat and, skimmed). For example, whole milk has bioavailable vitamin A, retinol. This analysis shows that, among the species studied, Bactrian camels, buffaloes, sheep and cows have high concentrations of vitamin A in 100 g of milk (Table 2). Sheep, cow and buffalo have high vitamin B12 content per 100 g of milk (Table 2). The highest levels of iron per 100 g of milk are in yaks and dromedary (Table 2). The highest zinc concentrations per 100 g of milk are in llama and reindeer, followed by yak and Bactrian camel (Table 2). Raw cow’s milk contains relatively low levels of vitamin D in comparison to other TASFs. However, commercially available products may be fortified with additional vitamin D. Regarding water-soluble vitamins, milk contains high levels of vitamin B complex and some vitamin C, which vary among livestock species.

Camelid milk contributes to food security and nutrition, providing high-quality food, especially to communities living in arid and semi-arid regions [25]. A study analyzing the milk nutrient concentrations of camelids such as dromedary and Bactrian camel revealed that, on average, they contain high levels of vitamin C (up to ten times more than cow’s milk) and high concentrations of total salts, calcium and other minerals, such as iron, copper and zinc, while exhibiting low levels of cholesterol [26]. Camel milk does not contain β-lactoglobulin, which is a protein found in cow milk that causes allergy [27].

One review indicated that non-cow milks may be easier to digest than cow’s milk, likely due to the formation of softer curds in the stomachs of humans during gastric digestion [21]. This effect might be attributed to the differing composition of milks by livestock species, including variations in casein compounds, fat globules and protein-to-fat ratio. Another study comparing various animal milks, including human milk, and found that fat content showed the most variation compared to other nutrients. In addition, human milk shares more content similarities with non-ruminant milks than with ruminant ones [28]. For instance, the structure of fat globules and triacylglycerol in ruminant milk (cattle, goat, sheep and buffalo) differed significantly from that in non-ruminant (donkey, horse and human) milk. Moreover, higher levels of unsaturated fats and lower saturated fats and MUFA were found in non-ruminant milk than in the ruminant one [28]. A different study comparing milks among ruminant species (buffalo and cow) revealed that the former had higher concentrations of protein, fat, calcium and vitamins A and C but lower levels of cholesterol, riboflavin and vitamin E [29]. Additionally, an in vitro experiment showed that fermented buffalo milk demonstrated higher bacterial viability than cow’s milk [30].

### 3.3. Unprocessed Meat

As with other TASFs, unprocessed meat contains essential nutrients that can contribute to human nutrition and health. However, the nutrient content varies, among other things, depending on animal-by-animal species and between meat cuts and leanness. Meat, similar to eggs and milk, is an unequivocal source of high-quality proteins. Meat from muscle tissue encompasses a variety of amino acids, including all the essential ones, and has high digestibility results in the Protein Digestibility—Corrected Amino Acid Scores. However, this score is lower than that of eggs [31]. The most prevalent amino acids are glutamic acid and glutamine, followed by arginine, alanine and aspartic acid [32]. Meat from poultry is noted for having low concentrations of structural protein collagen, which enhances digestion [33]. Meat with the highest protein content per 100 g in our analysis is found in pheasant, turkey and rabbit meats (Table 3).

Some of the fatty acids contained in meat include oleic (C18:1), palmitic (C16:0) and stearic (C18:0) acids [34]. Meat is also one of the primary dietary sources of docosapentaenoic acid (DPA) (C22:5 omega-3), which is available from mammal and poultry meat [35]. Additionally, it can also supply the long-chain fatty acids DHA and eicosapentaenoic acid (EPA), which are significant for human health [36]. According to a study, removing poultry skin can make poultry meat serve as a valuable source of unsaturated fats [33]. The diet of the animals has a greater impact on the fatty acid content in the meat of monogastrics than in that of ruminants [2]. This is due to how fatty acids are metabolized in the rumen (one of the four compartments of the ruminant stomach), which is absent in monogastric animals. The fermentation, lipolysis and biohydrogenation processes in the rumen influence meat from ruminants, which are rich in conjugated linoleic acid and unique branched-chain fatty acids that have benefits in human health [37,38]. However, it usually has greater levels of saturated fats in comparison with monogastrics [39,40]. Cooking methods significantly affect the content of fat and fatty acids in meat [31]. A study analyzing meat from beef and lamb revealed that cooking increased the content of omega-3 and omega-6 polyunsaturated fats [41]. Among livestock species, meat from pigs has the highest content of fat per 100 g, followed by sheep and cattle (Table 3).

Similar to other types of TASFs, meat supplies vitamin B12 along with other B vitamins—vitamin B3 (niacin), vitamin B2 (riboflavin) and vitamin B5 (pantothenic acid). Poultry meats are particularly noted for their vitamin B1 (thiamine), vitamin B6 and vitamin B5 contents. It has been observed that cooking techniques impact vitamin losses in meat, with vitamin B12 showing more significant decreases in comparison to other B vitamins [31,42,43]. All organ meats, with the exception of tripe, are good sources of vitamin B12 [32]. The liver contains the highest levels of choline and DHA [44]. It is also rich in the bioavailable form of vitamin A, retinol, along with iron and folate. Meat provides the vitamin D metabolite 25 hydroxycholecalciferol, which has been shown to have high biological activity [45].

Meat is a key dietary source of iron and zinc, which are the critical limiting minerals in diet globally. Meat commonly contains iron complexed as haem iron, which is absorbed in higher rates (on average 25%; range 15–35%) than the non-haem iron primarily located in plant-source foods (2–3%) [46]. Previous studies have shown similar findings, with rates ranging from 15% to 25% for haem iron and from 5% to 12% for non-haem iron [47,48,49]. In addition, the bioavailability of zinc found in meat is higher compared to that of plant-source foods [46]. Some other minerals contained in meat are selenium, copper and phosphorous. Fresh or unprocessed meat is low in sodium.

When comparing the mineral content in unprocessed meats, our analysis shows that the highest levels of iron per 100 g are found in goat and horse and in cattle and pigeon meats (Table 3). For zinc, the meat of goat, horse, cattle and goose has the highest content per 100 g (Table 3). Where vitamins are concerned, vitamin A is particularly high in 100 g of meat from pheasant and pigeon, followed by sheep and goat (Table 3). The highest levels of vitamin B12 per 100 g among the species compared are found in the meat of deer and rabbit (Table 3).

The bioactive compounds present in meat, carnitine, creatine, taurine, hydroxyproline and anserine, have been demonstrated to confer benefits to human health [50]. Taurine, primarily obtained from meat, functions as an antioxidant, along with other important roles in human health [32]. Evidence suggests that the concentration and metabolism of carnitine rise during pregnancy and lactation, and it might be crucial in infants during the first 1000-day period [51].

Other studies have examined the micronutrient composition of different meats. One review which covers various meats consumed in Australia and New Zealand found that mutton was more nutrient-dense than beef, veal and lamb [32]. Other recent reviews have also explored the characteristics of some meats that are less consumed, including processed sheep and goat meat [52]; pheasant, quail and guinea fowl [53]; and South American camelids [54].

Offal, which comprises viscera and blood, has substantial nutritional value, including significant amounts of high-quality proteins, vitamin A, vitamin B12, iron, zinc and essential fatty acids [55,56]. However, due to several factors, such as culture and consumer preferences, its consumption remains limited in some parts of the world [55,57]. The liver and kidney have shown high nutrient density. In particular, for protein, the liver has the highest content, while the intestines have the lowest [55]. Low fat content is found in the liver, heart, kidneys and lungs, while high levels of fat are found in the tongues of sheep, lambs and pigs [55]. In offal from lamb and sheep, saturated fatty acids account for 45–70% of their fatty acids [55,58]. Offal’s high levels of unsaturated fatty acid make it prone to oxidation during heat treatment and storage [55,59]. The high levels of vitamin B12 in ruminants are attributed to the biosynthesis of cobalamin by the bacteria and archaebacteria inhabiting their digestive system [55,60]. Additionally, iron is present in substantial amounts in the liver [55,56].

### 3.4. Food from Hunting and Wildlife Farming

Over the past twenty years, there has been an increase in the consumption of meat from hunted animals (game) and meat from farmed wild animals [10]. The Food and Agriculture Organization of the United Nations (FAO) defines wild meat as meat from “terrestrial animal wildlife used for food” [61]. The European Union defines wild game under European Union Regulation No. 853/2004 as “wild ungulates and lagomorphs, as well as other land mammals that are hunted for human consumption and are considered to be wild game under the applicable law in the Member State concerned, including mammals living in enclosed territory under conditions of freedom similar to those of wild game, and wild birds that are hunted for human consumption”. There is a wide range of hunted animals across multiple taxonomic groups, including bats, birds, hares and rabbits, kangaroos, rodents, reptiles, and ungulates.

Historically, hunting was a common practice, especially among people living in rural areas. There has been a growing demand for wild meats in high-income countries as they are seen as healthier, leaner and free of antibiotics and hormones. In some low- and middle-income countries, hunting and wildlife farming are crucial for food security [62]. One study identified 15 countries that could face significant food insecurity if prohibitions were introduced [63]. In South America, it was estimated that between 5 million and 8 million people frequently relied on meat from wild animals as a source of protein in the early 2000s [64].

Numerous authors have highlighted that meat from hunting and wildlife farming has lower levels of fats and cholesterol, along with higher levels of proteins, essential fatty acids, vitamins and minerals than meat from livestock meat [65,66,67]. One review of meat from wild animals analyzed the differences in nutrient composition among representative species [10]. While total protein concentrations do not vary significantly, some types of meat exhibit higher average concentrations. For example, meat from the common duiker (*Sylvicapra grimmia*), hare (*Lepus europaeus*), wild rabbit (*Oryctolagus cuniculus*), elk (*Alces alces*), roe deer (*Capreolus capreolus*) and fallow deer (*Dama dama*) have the highest concentrations [10]. In addition, some species have markedly higher total fat concentrations: the pigeon (*Columba livia*), springbok and fallow deer (*Dama dama*) [10].

Several studies have explored the nutrient profiles among various meats from wild animals or compared them to meat from livestock species. A nutritional analysis of meats from wild European animals, such as red and fallow deer, wild boar, hare and wild rabbit, highlighted that these meats have high protein and low fat levels [68]. In comparison to meat from livestock species, wild meats were identified to have higher levels of omega-3 PUFA and overall PUFA, and wild ruminants exhibited lower omega-6/omega-3 ratios (around 4 on average). These meats from wild European animals were noted for their high nutrient densities, including phosphorous, potassium, zinc, iron, vitamin E and the vitamin B complex. A different study, which analyzed meat from red and fallow deer with that from Aberdeen Angus and Holstein cattle, detected lower total concentrations of crude fat and collagen, as well as higher PUFA to saturated fatty acid ratios in the venison than in the beef [69]. The atherogenic index, measuring the combination of triglycerides and high-density lipoprotein cholesterol, was also found to be lower in venison compared to beef. An analysis of the fatty-acid profiles of some wild North American and African ruminants found them to be similar to those of pasture-fed cattle but different from those of grain-fed cattle [70].

Further studies have assessed the nutrient concentrations of meat from wild animals focusing on certain regions: wild fallow deer (*Dama dama*) in South Africa [71]; wild axis deer (*Axis axis*) in Croatia [72]; wild boar in Italy [73,74]; birds and game hunted by the Eastern James Bay Cree people of Quebec, Canada [75]; European game meat species [76]; wild boar in Latvia [77]; wild ungulates in Italy [78]; and wild animals in Nigeria [79]. The results from these studies are consistent with the reviews described in the whole section where, generally, meat from wild animal species shows high levels of high-quality proteins and fats and a range of several micronutrients.

### 3.5. Insects

The practice of consuming insects, also known as entomophagy (from Greek *éntomon*, ‘insect’, and *phagein*, ‘to eat’), has played a significant role in human evolution and is still prevalent in many cultures [80]. For example, in the northern regions of Thailand, insects are readily available as meat substitutes [81], and in Mexico, consuming insects has been linked to cultural traditions [82]. A study revealed that over 1900 insect species are globally edible [83], with beetles (Coleoptera) (31%) leading the list followed by caterpillars (Lepidoptera) (18%); bees; wasps and ants (Hymenoptera) (14%); grasshoppers; locusts and crickets (Orthoptera) (13%); and cicadas, leafhoppers, planthoppers, scale insects and true bugs (Hemiptera) (10%). Another analysis identified 2205 species as edible, with Asia being the region with the most (932 species) and Mexico as the country with the highest number (450) [84]. They have not only been a part of human diets for centuries but also have held significance in sociocultural and religious practices. Insects are gaining increasing recognition for their nutritional value and potential benefits in human health, supporting environmental sustainability and local livelihoods [85]. However, the literature sheds light on the current challenges of the consumption and acceptance of insects and insect-based products, which include neophobia (fear or aversion to new things) and disgust [86,87].

Insect’s nutritional content can differ based on species, stage of metamorphosis, sex, habitat, and diet [88,89]. For instance, larval and pupal stages generally have greater calories and fat levels than adult insects, and female insects typically showcase higher fat concentrations and energy levels than their male counterparts.

A systematic review of edible insects globally emphasized the wide range of nutrient content found in insects [90]. Studies have demonstrated that insects are rich in proteins and healthy fats. A review evaluating insect nutrient composition showed that some insect species have higher protein, PUFA and cholesterol concentrations compared to other TASFs [91]. However, these contain lower levels of saturated and MUFAs, thiamine, niacin, cobalamin and iron. Similarly, another review that assessed insects as a source of dietary protein noted that the order Blattodea (cockroaches) possessed the highest average protein content in comparison to the orders analyzed [92]. Findings from a review analyzing 236 species indicated that edible insects could provide enough protein, MUFA and/or PUFA to fulfill daily nutritional requirements [93]. A more recent study explored the viability of insect farming for both human food and animal feed, assessing the macronutrients of ten commonly used insects [94]. It concluded that protein level in the orders Orthoptera and Diptera (flies) oscillates between 51% and 76%. Fat concentration was consistently lower than protein across all orders, with some species exceptions, including *Galleria mellonella* (the greater wax moth) (51.4–58.6) [94]. In a study examining 212 insect species from Africa, investigators found that Lepidoptera had the highest protein content (20–80%) and fat content (10–50%), whereas Coleoptera, being the most consumed order, had the highest carbohydrate content (7–54%) [95].

Regarding micronutrients, some studies have reviewed the composition of different insects. The same systematic review mentioned above analyzed data from 91 species and found that among minerals, potassium levels were especially high across multiple species. In contrast, copper concentrations were relatively low. One study found that the content of these minerals across commonly reared and wild-harvested insects was close to or higher than that of other TASFs [96]. The authors, however, observed that the iron and zinc in insects originate from non-haem compounds, making their bioavailability unknown. Some studies have suggested that insects could potentially help mitigate iron and zinc deficiencies globally [96]. Regarding vitamins, insects were observed to have high concentrations of vitamin E and low concentrations of vitamin C. The vitamin C and dietary fiber found in insects have also been noted as a possible source of health benefits [91]. Insects also contain several bioactive compounds such as antioxidant peptides, polyphenols and flavonoids [86,97].

Insect powders and processed products obtained through drying and fermentation, along with other processing methods, are being considered for use for their potential to meet nutritional needs and alleviate malnutrition globally [98,99].

## 4. Differences in TASF Nutritional Quality, by Animal Characteristics and Livestock Production Systems Characteristics

Below, we summarize the findings of our analyses on how animal characteristics and production systems influence the quality of TASFs. We divided the section by intrinsic animal characteristics (genetic traits and non-genetic differences) and aspects of livestock production systems (feed, environmental conditions and climatic zone, housing conditions and other husbandry practices). To facilitate a comparison with the previous section, Table 4 and Table 5 illustrate the impact of genetic traits and husbandry practices on meat, milk and egg quality, respectively. Meat, milk and eggs are the main TASFs that we focus on in the section due to the limited existing evidence on other TASFs. For example, there was limited information on how variations in the breed and animal characteristics impact the nutrient content of organ meat. A review summarizes the available information on the determinants influencing the nutritional quality of offal [55]. We did not include information on food from hunting and wildlife farming in this section. However, the literature indicates that there are many factors that influence the nutrient composition of meat from hunting and wildlife farming, such as the season of the year, the climatic conditions and environment, sex and age [100].

### 4.1. Intrinsic Characteristics of Animals Impacting the Nutritional Quality

#### 4.1.1. Impact of Genetic Traits

With the exception of chicken and, to a lesser extent, milk, genetics has been reported as having a minor impact on the protein content of TASFs for most livestock species [101,102]. For example, for milk, some breeds examined show a higher content of protein than others. This is the case of the milk of Alpine goats, which presents a higher protein concentration than that of Saanen goats, and similarly, the milk of Norman and Jersey cattle contains more in comparison to Prim’Holstein cattle [110,111]. Conversely, the variation in fat content and fatty acid profile is more pronounced both among and within a breed. Broiler breeds with slower growth rates tend to accumulate higher fat content and have higher levels of total PUFA content in their meat [103,104]. Additionally, products from animals of locally adapted breeds often exhibit different nutrient content compared to those from international transboundary breeds that are subject to intensive genetic improvement programs. For example, meat from some local breeds of sheep [109], duck [105] and pig [107] is typically higher in PUFA, especially omega-3, and has lower omega 6 to omega 3 fatty acid ratios, which are considered more favorable for human nutrition compared to meat from international breeds. Ibérico meat, for instance, showed less than a half omega-6 to omega-3 ratio, PUFA and linoleic acid content compared to Berkshire meat. The impact of breed on nutrient content should be assessed in relation to feeding systems and the geographical area where the animals are raised [155,156].

Certain genes directly associated with quality traits are targeted in genetic improvement programs aimed at enhancing the commercial quality of TASFs. A prominent example is a mutation of the myostatin gene, which can be associated with “doubled-muscle” animals. These animals present muscle hypertrophy (on average 20% heavier), lean carcass and more tender meat, lower total fat, approximately 50% lower levels of saturated fatty acids and higher PUFA levels [157,158,159].

#### 4.1.2. Impact of Non-Genetic Differences

Non-genetic differences among individual animals, such as the age and sex of the animal, can also affect the nutritional quality of TASFs. The lipid content of meat products can be influenced by the sex of the animal and thus impact both the nutritional and the organoleptic quality. Females and castrated males typically have higher levels of intramuscular fat. The protein and fat content and the fatty-acid profile of an animal carcass are influenced by the age of the animal at slaughter. In particular, the fat content in meat generally rises as the animal ages. Genetic selection for growth rate and carcass yield in chickens has resulted in a reduced slaughter age and, thus, a higher moisture-to-protein content ratio in the standard retail product [103].

### 4.2. Livestock Husbandry Practices and Other Characteristics of Production Systems Impacting the Nutritional Quality

The quality of TASFs is significantly influenced by livestock management practices and production system characteristics. The method chosen to raise an animal can have a significant impact on the quality of the resulting TASFs. Livestock production systems vary significantly based on climate and other geographical factors, as well as husbandry practices, feed and feeding systems, housing and animal welfare conditions. For example, camel milk (from Bactrian camel and dromedary) tends to vary more in composition than cow milk depending on the feeding system, breed, age and lactation stage of the animal [27]. The animal’s diet is the primary factor influencing the nutritional quality of the resulting TASFs. Below, we describe three key characteristics of the livestock production systems that may have an impact on the quality of TASFs: feed and feeding systems, environmental conditions and climatic zones, and housing conditions, along with other husbandry practices employed.

#### 4.2.1. Feed and Feeding Systems

Feed types, including grass, forage, cereals or other concentrate feed, significantly impact the nutrient content, particularly the fatty-acid and vitamin profile of eggs, milk and meat. A diet rich in PUFA improves the levels of essential fatty acids in milk and meat. The omega-6 to omega-3 ratio is influenced more by the diet of the animals than by their genetic characteristics. Differences in the digestive systems of ruminants (cattle, buffalo, sheep, goat, deer) and monogastrics (pig, horse, rabbit and poultry species) result in varied fat absorption and digestion [160]. In monogastric animals, dietary fatty acids are absorbed directly without undergoing biochemical reactions in the stomach, and their composition is reflected in that of the meat. Conversely, ruminants undergo ruminal biohydrogenation of fatty acids, which reduces their absorption but allows the synthesis in the rumen of conjugated linoleic acid, a nutritionally beneficial fatty acid. This is correlated with the consumption of forages, which is a major part of the diet of ruminants. Moreover, monogastric animals (including rabbits and horses) can transform alpha-linolenic acid into PUFA. Nevertheless, the conversion rate is not very high. Some feeding strategies, such as grazing and omega-3-rich feed supplements (e.g., linseed), boost the levels of PUFA in milk and meat, resulting in a more nutritionally beneficial fatty-acid profile for human consumption. The positive influences of grazing are especially pronounced when animals consume leguminous pastures or forages from varied grasslands [121,161,162]. Grasses contain higher levels of alpha-linolenic acid, an essential fatty acid and precursor of omega-3 fatty acids, whereas cereal grains are rich in linoleic acid, a precursor of omega-6 fatty acids [163,164]. Consuming linseed in the diet, particularly as oil rather than whole seeds [162], enhances alpha-linolenic acid levels in the meat of cattle, sheep, pigs, rabbits and chicken [125,130].

In Brazil, a study analyzing milk from dairy goats raised in different feeding systems showed that goats that were raised on semi-arid native pasture exhibited a better milk fatty-acid profile in comparison to goats kept in a confined system: the native pasture consisted of 71 different plant species, while the confined-feeding system was derived from hay produced from Napier grass (*Pennisetum purpureum*) [135,136,137]. The quality traits and fatty acid profile of beef are also influenced by feeding systems. In Uruguay, beef from animals finished on pastures contained approximately twice the concentration of conjugated-linoleic acid than that from animals that finished in feedlots, regardless of whether they were raised in a feedlot or on pasture [119].

The use of insects in animal feed has shown a non-negligible impact on the nutritional quality of TASFs. Specifically, the inclusion of black soldier fly and yellow mealworm in the diet of swine and broilers has been found to change the fatty acid profile of the meat [165]. However, further research is needed to explore the linkages.

Using plant by-products that cannot be eaten by humans instead of human-edible feed crops helps lessen the environmental effects of livestock production. A review documents the environmental consequences and the influence on the nutritional quality of meat using a wide variety of different plant by-products as feed [131]. Including as much as 30% of distillers’ grains with soluble (maize, wheat) or glycerine in animals’ feeding has been linked with an increase in the PUFA levels of beef and lamb [131]. The inclusion of citrus pulp in the diet of lambs does not influence performance or carcass or meat quality and reduces rumen biohydrogenation of PUFAs along with lipid and protein oxidation [131]. Other fruit extracts are noteworthy for their role as natural antioxidants in animal diets. A recent meta-analysis indicated that, besides reducing greenhouse gas emissions, tannins supplementation in sheep resulted in their meat having higher levels of beneficial fatty acids (14% increase in omega-3) [132]. Overall, grazing animals have lower milk yield (21%) but higher protein levels (8%) in their milk compared to animals fed on a diet consisting of grass silage, maize silage and concentrate [166]. Increased concentrations of vitamin A and E have been reported in milk and meat of ruminants raised on a grass-based diet [138,167], and higher vitamin E concentrations in meat from free-range chickens [103]. Beef from pasture-fed systems has been shown to contain precursors of vitamin E and vitamin A at levels up to four times and seven times higher, respectively, than those found in concentrate-fed systems [133,134]. The literature reports no variation in the protein content of eggs and meat.

#### 4.2.2. Environmental Conditions and Climatic Zone

Excluding some vitamin and mineral supplements, the diets of livestock are primarily plant-based. Environmental characteristics such as climatic conditions, soil type, altitude and season influence the chemical composition of plants and, accordingly, livestock feed and, consequently, the food quality of the subsequent TASFs. Temperature and day length influence plant growing season, which in turn affects the accessibility to pasture and the quantity and quality of the forage for grazing animals. Compared to milk produced in other seasons, milk produced during the summer is characterized by higher levels of vitamin D and PUFA and lower levels of fat. A study based in Australia revealed that while latitude did not impact the levels of vitamin D3 in lean beef, fat from cattle grown in low-latitude environments had a higher content of vitamin D3 than fat from cattle raised in high-latitude regions [144]. Some studies highlight that the milk and meat of sheep, cattle and yaks raised on high-altitude pastures have high-quality fatty-acid levels [145,147]. The fatty-acid profile of TASFs is connected to the climatic zone and feeding system, specifically the combination of pasture management, grass maturity, grass diversity, season and altitude, which impacts the levels of alpha-linolenic acid (precursor of long-chain omega-3) in the animal’s diet [168].

#### 4.2.3. Housing Conditions and Other Husbandry Practices

Husbandry practices involve all the methods and instruments employed by farmers and pastoralists to manage animals until the time of slaughter. These include feeding, housing, reproductive management, and health and welfare care. The nutritional contents of TASFs are often positively influenced by production systems that incorporate grazing or free-range feeding [169]. Ducks raised in irrigated rice fields in China have been observed to exhibit higher carcass weight and intramuscular fat, lower protein content and higher concentrations of some essential amino acids (valine, methionine, phenylalanine, histidine and arginine) and PUFA than ducks raised in floor pens [128]. Since vitamin D is synthesized through sunlight exposure of animals, housing and indoor rearing reduce the levels of vitamin D in cow milk [154].

Organic production is on the rise within the livestock sector as awareness around animal welfare and environmental sustainability continues to increase among both producers and consumers. The absence of international common standards for organic production hampers comparisons among those production systems as well as the assessment of their influence on the nutritional quality of TASFs. Variations in study results arise from variations in husbandry practices and combinations of circumstances, including diet, housing and breed, and age of the animals. Nevertheless, aims have been made to assess the impact of organic production systems on the nutritional quality of TASFs. Two meta-analyses described that specific organic meats (beef, lamb and pork) and cow milk have healthier fatty-acid profiles than their non-organic equivalents. This means higher levels of PUFA in all TASFs studied and higher conjugated linoleic acid and vitamin E levels in milk [126,127]. These results may be closely related to pasture access and grazing and forage-based diets specified under organic production standards [126]. As an example, a study noted that the omega-6/omega-3 ratio was 2.5 times higher in conventional milk than in organic one. An increase in this dietary ratio might lead to potential health concerns [170,171].

## 5. Discussion

Our study found variation and contrast in the nutritional quality of TASFs by livestock species and animal products, as well as by characteristics of livestock production systems. We compiled nutrient data from six geographically diverse food composition databases and synthesized the literature to evaluate five TASF groups—poultry eggs, milk, unprocessed meat, foods from hunting and wildlife farming, and insects. The analysis particularly focused on essential nutrients, including vitamin A, iron and zinc, which are often limiting nutrients, particularly in low-and middle-income countries. We included insects and foods from hunting and wildlife farming, often lacking evidence on animal source foods. Findings emerged linking livestock species to specific nutrients with human health implications, highlighting variations in nutrient content among species from lowest to highest quartiles. For example, quail and goose eggs, reindeer milk, pheasant meat, meat from hunting and wildlife farming species of common duiker, hare, wild rabbit, elk, roe deer and fallow deer [10], and the insect order Blattodea [92] showed the highest levels of protein by the TASF groups analyzed. We also uniquely summarized the evidence base for intrinsic animal characteristics and characteristics of livestock production systems. We found more literature on the influence of fatty acids in comparison to other macronutrients, particularly highlighting the significant impact of feed and grazing systems on cattle.

The findings confirm that TASFs are an important source of the nutrients analyzed, which, in most cases, are limiting in vulnerable populations and influence nutrition and health outcomes. We found that meat, including from wildlife farming species, has the highest quantity of protein of the TASFs analyzed per 100 g. Due to the combination of amino acids found in eggs, they are often used as a benchmark against which other proteins are compared [13]. On average, protein content in eggs is relatively consistent among poultry species (Table 2). Reindeer have the highest content of protein per 100 g in their milk, followed by mithan, sheep and alpaca (Table 1). In general, consuming high-quality proteins, which are often found in TASFs, enhances human health outcomes; for example, it can promote the growth of muscle mass throughout the life course [172]. During older adulthood, higher intakes of protein are important to prevent muscle loss (sarcopenia) [173,174]. Over the past few years, an increased emphasis on the significance of high-quality proteins in preventing child-stunted growth has been observed [175,176]. Some amino acids, as well as their metabolites, are notably found in TASFs and either absent or present at low levels in plant-source foods. For example, some TASFs may contain higher concentrations of tryptophan, a precursor of serotonin, which has been recently found to be linked to brain function via the microbiome [177], as well as to depression in older adults [178] and postnatal women [179]. Two systematic reviews concluded that insect protein is a suitable protein source with promising health benefits [180,181].

Feeding strategies, livestock production systems and types of supplementation could significantly influence the content of fatty acids such as omega-3 and omega-6 in meat and milk [182]. According to studies, insects exhibit a similar fatty acid composition to poultry and fish but higher levels of unsaturated fatty acids, particularly PUFA [183,184]. The profile of essential fatty acids found in TASFs influences human health outcomes. Specifically, the ratio of linoleic acid to alpha-linolenic acid impacts the efficiency with which these fatty acids are endogenously converted into the longer chain fatty acids arachidonic acid, aminopenicillanic acid and docosahexaenoic acid (DHA) [185]. Maintaining a balanced and low omega-6/omega-3 ratio is crucial for the synthesis of long-chain fatty acids. Long-chain fatty acids are recognized for their positive impact on human health and play critical roles throughout the life course, including inter alia in neurodevelopment, anti-inflammatory processes and cell-membrane integrity [186,187]. The current review identified that chicken eggs contain fewer lipids than the other poultry species analyzed (duck, goose, quail and turkey) (Table 1). Fat content per 100 g of milk is high in reindeer, followed by mithan, buffalo and yak. The species with the lowest protein and fat content are mare and donkey (Table 2). Sphingomyelin, which is found in animal milk, may help protect against gut dysbiosis and inflammation [188]. The fat content per 100 g of meat is the highest in pork (37.2 g), followed by sheep and cattle meat (34 g) (Table 3). Meat serves as one of the primary dietary sources of docosapentaenoic acid (DPA) (C22:5 omega 3), which is primarily available from mammal and poultry meat [35]. The literature indicates that DPA likely may help lower the risk of chronic disease [189].

Lactose is found in the composition of milk and dairy products. This varies among animal species and food products. Some people are unable to digest the disaccharide, which can cause allergies or sensitivity. Tolerance or sensitization of cow’s milk is affected by various factors, including genetic predisposition; infections; alteration of intestinal microflora; age at first exposure; maternal diet; antigen transmission through breastmilk; and the nature, quantity and frequency of antigen load [190].

In our analysis, duck eggs showed the highest concentration of vitamin A expressed in retinol equivalents (Table 1). Eggs are recognized for containing carotenoids such as lutein and zeaxanthin, which play a crucial role in anti-inflammatory pathways [191,192]. Vitamin A, a fat-soluble vitamin, is important for normal vision, the immune system, reproduction, and growth and development. In TASFs, it is found in preformed vitamin A, which is more bioavailable than in some plant-sourced foods in which it is found in provitamin A carotenoids [193]. There has been a rising focus on choline in recent years. Choline acts as a precursor for phospholipids (phosphatidylcholine and sphingomyelin), which are integral to cell-membrane integrity and signaling. It also contributes to the synthesis of acetylcholine, which influences neurotransmission, neurogenesis, myelination and synapse formation, and for betaine, which provides a methyl group in the homocysteine production pathway [194,195,196,197]. Recent systematic and narrative reviews point to the significance of choline for growth and neurodevelopment in the first 1000 days of life [198,199,200]. Choline is found in high levels in eggs and, to a lesser degree, in beef. A randomized controlled study concluded that the early introduction of eggs as complementary feeding could increase choline pathway biomarkers [201]. The highest levels of vitamin B12 per 100 g of meat in our analysis are found in deer and rabbit (Table 3) and in the milk of sheep, cows and buffaloes (Table 2). A significant nutrient for human health mostly found in TASFs is vitamin B12 or cobalamin. It can also be found in seaweed, mushrooms and tempeh, although in forms that may not be as bioavailable as the vitamin B12 in TASFs [202,203]. Cobalamin is crucial for cellular metabolic processes such as DNA synthesis and methylation. A deficit in vitamin B12 may lead to pernicious anemia and compromised neurodevelopment and cognitive function [204,205].

Turkey eggs are richer in certain key minerals than eggs from other poultry species in some important minerals, comprising calcium, iron and zinc, followed by quail eggs (Table 1). The minerals zinc and iron are found to be highly bioavailable in the muscle tissue of meats. For zinc, the highest levels per 100 g among the species analyzed in the study are found in goat, horse, cattle and goose meats (Table 3). For milk, the highest zinc concentrations per 100 g are contained in llama and reindeer, yak and Bactrian camel (Table 2). Zinc has an important role in the activity of over 300 enzymes in the human body and also plays essential roles in growth, development and immunity [46]. Animal milk provides complementary nutrients that synergistically optimize metabolism, for example, lactose and casein acting to enhance calcium absorption [22]. When comparing the mineral concentrations of various meats, it shows that the highest levels of iron per 100 g of meat are found in goat and horse and in cattle and pigeon meats (Table 3). The species with the highest levels of iron per 100 g of milk are yaks and dromedaries (Table 2). Iron plays multiple roles in the human body; it is noted for its role in oxygen transportation in the hemoglobin blood protein, but it is also involved in other pathways of growth, neurodevelopment and immunity [46,206]. Insufficient levels of iron and zinc can substantially influence the global burden of disease [207].

The study confirms that, in certain settings, studying further applications of livestock production systems, for example, an in-depth analysis of the influence of genetic trait selection, feeding or animal welfare practices, can lead to the elevation of quality and sustainability of the production of TASFs. Heritable traits that influence nutrient composition have been incorporated into genetic selection programs. For example, a genetic selection program has targeted the A2 variant beta-casein in the milk of dairy cows to obtain hypoallergenic and more digestible milk, called A2 milk [106]. A2 milk has been commercialized in New Zealand and the United States of America. The rising demand for nutritional quality and other characteristics related to perceived sustainability has sparked commercial initiatives to promote TASF’s nutritional quality through on-farm production practices. For example, the “Bleu-Blanc-Coeur” supply chain uses cattle feed with a higher content of omega-3 to produce TASFs with a targeted fatty acid profile with higher omega-3 content. The supply chain model has been developed in different regions, including Europe, Northern and Southern America, and Asia.

It is essential to consider TASFs in the context of a healthy diet. Our analysis focused on data by single limiting nutrients. However, for human nutrition, it is crucial to consider the complexity of the food matrix, which influences the absorption and metabolism of nutrients from foods [208]. TASFs include bioactive compounds in their matrix. These are phytochemicals found in foods or by-products and are able to regulate metabolic functions, leading to favorable effects [209]. TASFs also include unique bioactive compounds that could be crucial for anti-inflammatory and immune functions, memory and cognition, and cardiovascular health, among other benefits [1,4]. Taurine, creatine, carnosine, 4-hydroxyproline and anserine have all been identified in cattle and are concentrated in beef but are absent in plant-source foods [50]. In addition, it is important to acknowledge that some TASFs include in their composition trans fats, which may derive naturally from ruminants (for example, from cattle, goats or sheep) or by hydrogenation in vegetable oil. Usually, evidence associates trans fats with negative health outcomes [210,211]. Nutrient adequacy, macronutrient balance, diversity and moderation are fundamental properties that best reflect healthy diets [212]. TASFs are one of the several food groups that can contribute to healthy diets and nutrition. The variability of nutrients across species from the analysis suggests the importance of incorporating and diversifying different kinds of animals and products, together with plant-source foods, into our diets.

The study has notable strengths, including the consideration of six databases geographically distributed, thus presenting a more comprehensive and global analysis. It included the nutrient review of insects and meat from hunting and wildlife farming, which are usually not considered when TASFs are an important part of the diets of some communities [213]; it focused on nutrients that are commonly lacking in low-and middle-income countries. This might present an opportunity to prioritize and identify high nutrient-density TASFs, considering the adaptability of various animal species and production practices to local contexts.

The extensive narrative review has found several gaps in the literature. Public health nutrition has generally concentrated on individual nutrients rather than adopting a broader perspective on the food matrix and dietary patterns influencing the digestibility of TASFs. Evidence from low- and middle-income countries regarding TASF nutrients is limited to some TASFs in food composition databases. While it could have important implications for human health, the existing research on food from hunting and wildlife farming and on insects remains limited. The review also had some constraints. Firstly, it did not provide an in-depth analysis of all TASFs, such as offal or dairy products, nor processing as it was out of scope. In addition, it did not cover all nutrients but the key ones that are comparable within and among TASFs, which have an important impact on health and nutrition outcomes. However, we acknowledge that TASFs provide other nutrients that also have an impact on nutrition and health outcomes, such as vitamin B6, which is highly bioavailable in meat [214]. We also recognize that TASFs are just one component of all the significant elements contributing to the overall diet quality. Other groups such as vegetables, fruits or cereals, even if, in some cases, they have a lower bioavailability of the analyzed nutrients, contain other essential nutrients important to human health, such as vitamin C. In addition, as a narrative review, some studies related to the main objectives would be missed. However, the research on the topic was performed extensively through three different databases to reduce this potential bias. Lastly, country and regional food composition databases were selected in order to have a global overview of TASF nutrient composition. Nevertheless, this can have a potential bias for some contexts and countries not included. Additional information on the construction of the tables can be found in Appendix A from the Appendix A.

## 6. Conclusions

TASFs are rich in high-quality proteins, fats and micronutrients, such as vitamin B12, iron or zinc, which impact human health. Their composition varies across livestock species, products and production systems. Extensive evidence emphasizes how certain husbandry characteristics influence the quality of fats in the TASFs but not on other nutrients. Our findings suggest that there may be public health opportunities by diversifying TASF consumption across species and improving certain aspects of the production systems to provide products that are both more sustainable and of higher quality. The Periodic Table of Foods Initiative might come as an opportunity to understand and make a more holistic analysis of how the composition of a food can impact nutrition and human health. The initiative addresses gaps in food biomolecular composition knowledge through a uniform, accessible and enabling platform based on analytical tools, data and capacity building [215]. Future research should adopt a more holistic approach to examining the food matrix and the dietary patterns that influence TASF digestibility. It is necessary to include meat from hunting and wildlife farming and insects in global food composition databases, as limited literature was found. In addition, scarce research focuses on low- and middle-income countries, highlighting the need for further exploration of TASF food composition analysis and how intrinsic animal characteristics and livestock production system characteristics impact their nutritional value.

## Figures and Tables

**Table 1 nutrients-16-03346-t001:** Nutrient differences across eggs from poultry species.

Nutrient ^1,2^	Energy (kcal)	Energy (kJ)	Protein (g)	Fat (g)	Carbohydrate (g)	Vitamin A (µg) RAE ^3^	Riboflavin (mg)	Vitamin B12 (µg)	Calcium (mg)	Iron (mg)	Zinc (mg)
Chicken (b, c, d)	140	586	13.1	9.6	0.4	137	0.47	1.2	53	2	1.3
Duck (a, c, d)	182	764	12.4	13.7	2.1	235	0.28	5.4	57	3.2	1.3
Goose (c)	185	775	13.9	13.3	1.4	187	0.38	5.1	60	3.6	1.3
Quail (c, d)	169	707	14.3	11.5	1.6	99	0.43	1.8	73	4.5	1.8
Turkey (c, d)	178	740	11.5	12.5	3.1	80	0.26	2.2	99	5.2	2.1

^1^ Shading is applied to the nutrient column, with varying intensities indicating different percentiles of the dataset: quartile 1 (light), quartile 2 (medium light), quartile 3 (medium dark), and quartile 4 (dark). Darker shade denotes higher nutrient average levels per 100 g of TASFs while lighter indicates lower average nutrient levels compared to other species from the table. ^2^ Nutrient value is expressed per 100 g edible portion on a fresh weight basis.^3^ Vitamin A content is expressed in retinol equivalents (RAE). (a) ASEAN Food Composition Database; (b) Australian Food Composition Database; (c) FoodData Central (USDA); (d) FAO/INFOODS Western Africa.

**Table 2 nutrients-16-03346-t002:** Nutrient differences across milk from mammalian livestock species and humans.

Nutrient ^1,2^	Energy (kcal)	Energy (kJ)	Protein (g)	Fat (g)	Carbohydrate (g)	Vitamin A (µg) RAE ^3^	Riboflavin (mg)	Vitamin B12 (µg)	Calcium (mg)	Iron (mg)	Zinc (mg)
Human (b, c, d, e, f)	71	294	1.2	4.1	7.3	62	0.04	0.05	31	0.07	0.2
Buffalo (g)	99	412	4.0	7.5	nd	69	0.11	0.45	191	0.17	0.5
Cattle (a, b, c, d, e, f,)	65	273	3.4	3.5	4.9	41	0.20	0.49	116	0.11	0.5
Mithan (g)	122	510	6.5	8.9	nd	nd	nd	nd	88	nd	nd
Yak (g)	100	417	5.2	6.8	nd	nd	nd	nd	129	0.57	0.9
Goat (a, c, d, e, f)	73	305	3.7	4.4	4.6	38	0.13	0.08	137	0.14	0.3
Sheep (c, e)	104	434	5.9	6.7	5.0	45	0.27	0.50	179	0.10	0.6
Alpaca (g)	71	299	5.8	3.2	5.1	nd	nd	nd	nd	nd	nd
Bactrian camel (g)	76	319	3.9	5.0	4.2	97	0.12	nd	154	nd	0.7
Dromedary (g)	56	234	3.1	3.2	nd	nd	0.06	nd	114	0.21	0.1
Llama (g)	78	326	4.1	4.2	nd	nd	nd	nd	195	nd	1.1
Reindeer (g)	196	819	10.4	16.1	nd	nd	nd	nd	320	nd	1.1
Donkey (g)	37	156	1.6	0.7	nd	nd	0.03	nd	91	nd	nd
Horse (g)	48	199	2.0	1.6	nd	nd	0.02	nd	95	0.10	0.2

^1^ Shading is applied to the nutrient column, with varying intensities indicating different percentiles of the dataset: quartile 1 (light), quartile 2 (medium light), quartile 3 (medium dark), and quartile 4 (dark). Darker shade denotes higher nutrient average levels per 100 g of TASFs, while lighter indicates lower average nutrient levels compared to other species from the table; nd indicates not available, presented in grey shade. ^2^ Nutrient value is expressed per 100 g edible portion on a fresh weight basis.^3^ Vitamin A content is expressed in retinol equivalents (RAE). (a) ASEAN Food Composition Database; (b) Australian Food Composition Database; (c) Tabla de Composición de Alimentos Colombianos; (d) Danish Food Composition; (e) FoodData Central (USDA); (f) FAO/INFOODS Western Africa; (g) [9].

**Table 3 nutrients-16-03346-t003:** Nutrient differences across meat from mammalian and avian livestock species.

Nutrient ^1,2^	Energy (kcal)	Energy (kJ)	Protein (g)	Fat (g)	Carbohydrate (g)	Vitamin A (µg) RAE ^3^	Riboflavin (mg)	Vitamin B12 (µg)	Calcium (mg)	Iron (mg)	Zinc (mg)
Cattle (b, c, d)	333	1388	16.5	34.0	0	0	0.08	2.73	246	3.48	2.3
Buffalo (c)	99	414	20.4	1.4	0	0	0.20	1.66	12	1.61	1.9
Sheep (a, b, d)	360	1807	14.7	34.0	0	32	0.11	1.99	8	1.22	1.3
Goat (b, c, d)	222	932	17.3	17.1	0	24	0.38	2.30	24	3.75	3.8
Pig (c, d)	392	1637	13.9	37.2	0	14	0.02	0.49	23	0.18	0.4
Horse (c)	133	556	21.4	4.6	0	0	0.10	3.00	6	3.82	2.9
Rabbit (a, b, c, d)	129	541	21.6	4.5	0	12	0.08	6.23	17	1.40	1.6
Deer (c)	120	502	23.0	2.4	0	0	0.48	6.31	5	3.40	2.1
Chicken (c, d)	180	756	18.9	11.4	0	27	0.11	0.25	101	1.16	1.3
Turkey (c)	115	479	22.6	19.3	0	9	0.19	1.24	11	0.86	1.8
Quail (b, c)	153	642	20.2	7.8	0	19	0.39	0.84	10	2.91	1.7
Pheasant (c)	133	556	23.6	3.6	0	50	0.15	0.84	13	1.15	1.0
Duck (a, b, c)	163	682	17.0	10.2	0	18	0.33	0.55	11	2.00	2.0
Goose (c)	161	674	22.8	7.1	0	12	0.38	0.49	13	2.57	2.3
Pigeon (b, c)	216.5	907	16.9	16.7	0	51	0.25	0.44	13	3.45	2.5
Guinea Fowl (c, d)	108	456	21.1	2.3	0	12	1.16	0.37	18	0.99	1.3

^1^ Shading is applied to the nutrient column, with varying intensities indicating different percentiles of the dataset: quartile 1 (light), quartile 2 (medium light), quartile 3 (medium dark), and quartile 4 (dark). Darker shade denotes higher nutrient average levels per 100 g of TASFs while lighter indicates lower average nutrient levels compared to other species from the table. ^2^ Nutrient value is expressed per 100 g edible portion on a fresh weight basis.^3^ Vitamin A content is expressed in retinol equivalents (RAE). (a) ASEAN Food Composition Database; (b) Australian Food Composition Database; (c) FoodData Central (USDA); (d) FAO/INFOODS Western Africa.

**Table 4 nutrients-16-03346-t004:** Animal characteristics impact on TASF quality: genetic trait.

TASF Product	Nutrients	Impact	Sources
Chicken meat	Protein and amino acids	A high breast–yield strain has a significantly lower protein content (3–4%) in comparison with a standard breast–yield hybrid and higher protein content in indigenous chickens.	[101,102]
		Genetic selection for growth has led to quality defects such as “white striping” or “wooden breast”, which are linked with a decrease in muscle content (7% to 18%) and an increase in collagen content (up to 11%).	[103]
Chicken meat	Fat and fatty acids content	Commercial hybrids with slower growth rates are generally fatter compared to cross-bred genotypes. Their meat generally has higher total omega-6 polyunsaturated fats (PUFA).	[103,104]
Duck meat		Native ducks had higher levels of PUFA, greater proportions of omega-6, enhanced nutritional value, an increase in the ratio of PUFA to saturated fatty acids (SFA) and lower content of SFA, atherogenic and thrombogenic indices compared to Peking ducks.	[105]
Pig meat		Meat from Ibérico pigs contained higher fat content (twice as much), a better fatty acid profile, with a 20% increase in oleic acid (monounsaturated fats (MUFA)), a 7% higher level of palmitic acid (saturated fatty acid) and nearly a 50% lower omega-6 to omega-3 fatty acids ratio, compared to meat from Landrace, Yorkshire and Duroc pigs, which had higher levels of saturated stearic acid.	[106]
		Native Korean pigs were found to have a 57% higher PUFA content compared to cross-bred pigs (Landrace x Yorkshire x Duroc).	[107]
Sheep meat		Tibetan sheep (an indigenous breed from high-altitude dryland) had a superior fatty acid profile compared to Small-tailed Han sheep (from northern China). Tibetan sheep had higher omega-3 PUFA, at least three-times lower omega-6/omega-3 PUFA ratio and higher concentrations of essential omega-3 fatty acids (alpha-linolenic acid, eicosapentaenoic acid (EPA) and docosahexaenoic acid (DHA)).	[108]
		Damara meat had greater levels of omega-3 fatty acids, including EPA, DPA and DHA.	[109]
Cattle, Goat, Sheep milk	Protein and fats content	Milk of Alpine goats’ fat and protein content is higher than in milk of Saanen goats and in the milk of Norman and Jersey cattle than in the milk of Prim’Holstein cattle.	[110,111]
Dromedary milk		The individual variability in milk composition (fat and protein) among animals of the same breed and physiological status, and when fed the same diet, is greater than the variation observed between different breeds.	[112]
Cattle milk	Vitamin	The beta-carotene content in milk was up to twice as high in Jersey cattle compared to Prim’Holsteins.	[110]
Chicken eggs	Protein and amino acids	When kept in the same environment and fed with the same diet, eggs from an indigenous breed had higher protein and higher yolk cysteine content than eggs from a hybrid population.	[113,114]
	Fat and fatty acids	When reared in the same environment with the same feeding program, SFA (14% more) and MUFA (50% more) are higher in eggs from a local Italian breed than in a hybrid breed.	[115]
		Eggs from a local Galician breed (Spanish breed) contained 26% more fat compared to eggs from the hybrid Isa Brown.	[113]
		There was no variation in the fatty acid profile between four local Portuguese breeds and a hybrid population.	[116]

**Table 5 nutrients-16-03346-t005:** Husbandry practices impact on TASF quality.

TASF Product	Characteristics of Production Systems and Agro-Ecological Conditions	Nutrients	Impact	
Cattle milk	Stage of lactation	Protein and fats content	Inverse relationship between milk quantity and the fat and protein content.	[110]
Donkey milk			Protein and casein content were higher in milk at the beginning of lactation.	[117]
Cattle meat	Feed and feeding systems	Fat and fatty acids content	A diet with a higher proportion of concentrates compared to forage is linked to a 20% increase in MUFA, a 13% decrease in SFA, a 73% reduction in omega-3 levels, and a fourfold increase in the omega-6/omega-3 ratio.	[118]
			In Uruguay, beef from animals finished on pastures was found to have approximately twice as much conjugated linoleic acid compared to beef from animals finished in feedlots, regardless of whether the animals were raised in a feedlot or on pasture.	[119]
Yak meat			A lower omega-6/omega-3 ratio was observed in meat from grazing yaks compared to meat from yaks feedlot-fattened after grazing pasture.	[120]
Cattle, Goat, Lama, Sheep meat			Feeding grass or forages that include omega-3 PUFA-rich plants increases the omega-3 content (including alpha-linolenic acid and EPA) by approximately in cattle and sheep and reduces the omega-6/omega-3 PUFA ratio by a factor of three to four.	[39,118,121,122,123,124,125]
Cattle, Sheep, Pig meat			Certain organic meats (beef, lamb and pork) and cow milk have healthier fatty-acid profiles compared to their non-organic counterparts.	[126,127]
Duck meat			Ducks raised in irrigated rice fields in China have been found to have higher carcass weight, with higher intramuscular fat, lower protein content and higher concentrations of some essential amino acids and PUFAs (omega-6 and omega-3), compared to ducks raised in floor pens.	[128]
Rabbit meat			The omega-3 PUFA content in rabbit meat doubled when the animals were fed a diet that included linseed.	[129]
			Caecotrophia in rabbits (reingestion of soft feces issued from bacterial fermentation in caecum) has been found to enhance PUFA content in the meat.	[130]
Sheep meat			Incorporation of citrus pulp in the diet of lambs had no impact on their performance, carcass or meat quality while limiting rumen biohydrogenation of PUFAs and reducing lipid and protein oxidation.	[131]
			Feeding tannins to sheep resulted in an increase in beneficial fatty acids in their meat, with omega-3 levels rising by 14%.	[132]
Cattle, Goat, Sheep meat		Vitamin	Grass-feeding results in a sevenfold in vitamin A levels, a 60% increase in vitamin C levels, and a twofold increase in vitamin E levels (or their precursors).	[133]
Cattle, Goat, Sheep meat			Grass-finished beef contains three times more vitamin B1, twice as much vitamin B2, and over three times the amount of vitamin E compared to grain-finished beef.	[134]
Cattle, Sheep, Pig meat; Cattle milk		Fat and fatty acids content	Certain organic meats, including beef, lamb and pork, as well as cow milk, have healthier fatty-acid profiles compared to their non-organic equivalents.	[126,127]
Goat milk			Milk from dairy goats raised on semi-arid native pastures demonstrated a better fatty-acid profile compared to that of goats kept in a confined system.	[135,136,137]
Cattle milk		Vitamin	Diets based on concentrate or maize silage lead to a 40% reduction in carotenoids and a 30% reduction in vitamin E content compared to grass-feeding.	[138]
Chicken egg		Fats and fatty acids content	When hens have access to pasture, there is a three- to fivefold increase in PUFA and a 50% reduction in the omega-6/omega-3 ratio, bringing it to around 5).	[139]
Quail egg			Seeds like hemp seeds increase omega-3 content in egg yolk, leading to a sevenfold increase in alpha-linolenic acid.	[140]
Chicken, Duck, Turkey egg		Protein and amino acid content	No variation.	[110,134,141,142,143]
Cattle meat	Environmental conditions and climatic zones	Vitamin	Latitude does not affect the vitamin D3 content in lean beef in Australia; however, fat from cattle raised at low latitudes had higher concentrations of vitamin D3 compared to fat from cattle raised at high latitudes.	[144]
Yak meat		Fat and fatty acids content	High altitude is associated with up to a 25% increase in the percentages of PUFA.	[145]
Cattle, Sheep milk	Plant composition of pasture	Protein and fat content	Grazing at higher elevations results in a more beneficial fatty-acid profile, with up to an 87% increase in PUFA concentrations, up to a 68% increase in omega-3 fatty acids (such as alpha-linolenic acid) and higher milk fat content.	[146,147,148]
Yak milk		Vitamin	Higher altitudes (ranging from 3 215 m to 5 410 m) are associated with a 60% increase in vitamin A content and a 28% increase in vitamin E content.	[149]
Yak meat	Season	Fat and fatty acids content	When pasture is abundant, there is an increase in fat content.	[150]
Cattle, Sheep milk		Vitamin	Spring and summer milk contains more vitamin D and calcium, less saturated fatty acids and more PUFA and omega-3 than winter milk.	[151,152,153,154]

## Data Availability

The original contributions presented in the study are included in the article and Appendix A; further inquiries can be directed to the corresponding authors.

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
