# Peer review of "Unveiling the Nutritional Quality of Terrestrial Animal Source Foods by Species and Characteristics of Livestock Systems"

_nutrients, 2024, doi:10.3390/nu16193346_

Round 1

Reviewer 1 Report

Comments and Suggestions for Authors

The manuscript presents a comprehensive literature review on TAFS, their potential as a source of nutrients in the human diet and the factors that could influence this potential. In addition to a detailed discussion of the results of the literature research, the authors have also outlined the strengths and limitations of the study. The authors have highlighted gaps in the literature on this topic and pointed out issues on which future research needs to focus. Overall, a well-prepared manuscript. There are only a few minor comments on my part:

Table 2. Add an explanation for the abbreviation 'nd'.

L216-217: The rumen is one of the four compartments of the ruminant stomach. Please correct the explanation in brackets.

L362: Please indicate which bioactive compounds this refers to.

Author Response

We greatly appreciate the comments from Reviewer 1. Please see our responses below.

Comment 1: Table 2. Add an explanation for the abbreviation 'nd'.

Response 1: The description is in the note below the tables. nd indicates not available, presented in grey shade.

Comment 2: L216-217: The rumen is one of the four compartments of the ruminant stomach. Please correct the explanation in brackets.

Response 2: We agree with the proposal and thanked the highlight. The sentence has been changed with the proposed sentence.

Comment 3: L362: Please indicate which bioactive compounds this refers to.

Response 3: Many thanks for pointing this out. We have added as an example, which is mentioned in the source cited, "antioxidant peptides".

Reviewer 2 Report

Comments and Suggestions for Authors

This paper reviews macronutrients, micronutrients and bioactive compounds in terrestrial animal source foods (TASF), focusing on five groups including poultry eggs, milk, unprocessed meat, foods from hunting and wildlife farming and insects.  This study is based on a literature search from three databases: EBSCOHost, PubMed and ScienceDirect. The nutrients of energy, protein, fat, carbohydrate, vitamin A, riboflavin (vitamin B2), cobalamin (vitamin B12), calcium, iron and zinc for TASF in the five groups are discussed. In general, this paper may be of interest to readers in this field.

1. Lines 583 and 584. “Meat is also the primary dietary source of docosapentaenoic acid (DPA) 583 (C22:5 omega 3), which is available from mammal and poultry meat but not from fish”. This may not be entirely correct. Some fish do contain DPA.

2. What does “nd” in Table 2 mean?

3.  In the discussion section, the comparisons of nutrients have been discussed. A table or figure can be added to provide a clearer overview of the ranking of TASF in terms of specific nutrient content.

4. In the insect section, no table shows nutrient differences like Tables 1-3.

Author Response

We thank Reviewer 2 for the comments and proposals for improvement. Please see our responses below.

Comment 1: Lines 583 and 584. “Meat is also the primary dietary source of docosapentaenoic acid (DPA) 583 (C22:5 omega 3), which is available from mammal and poultry meat but not from fish”. This may not be entirely correct. Some fish do contain DPA.

Response 1: I understand your point. I have changed the sentence to: “Meat is also one of the primary dietary source of docosapentaenoic acid (DPA) 583 (C22:5 omega 3), which is available from mammal and poultry meat".

Comment 2: What does “nd” in Table 2 mean?

Response 2: The description is in the note below the tables. nd indicates not available, presented in grey shade.

Comment 3: In the discussion section, the comparisons of nutrients have been discussed. A table or figure can be added to provide a clearer overview of the ranking of TASF in terms of specific nutrient content.

Response 3: Thank you for the idea, but we feel as though it may be repetitive to other tables already included in the manuscript (Table 1-3).

Comment 4: In the insect section, no table shows nutrient differences like Tables 1-3.

Response 4: Thanks for pointing this out. The databases used did not provide the information required to build up a table similar to tables 1 to 3 for insects.